# Legumain Functions as a Transient TrkB Sheddase

**DOI:** 10.3390/ijms24065394

**Published:** 2023-03-11

**Authors:** Christoph Holzner, Katharina Böttinger, Constantin Blöchl, Christian G. Huber, Sven O. Dahms, Elfriede Dall, Hans Brandstetter

**Affiliations:** Department of Biosciences and Medical Biology, University of Salzburg, Hellbrunner Str. 34, A-5020 Salzburg, Austria

**Keywords:** lysomale proteases, neurotrophins, tyrosine receptor kinase, functional processing, dimerization

## Abstract

While primarily found in endo-lysosomal compartments, the cysteine protease legumain can also translocate to the cell surface if stabilized by the interaction with the RGD-dependent integrin receptor αVβ3. Previously, it has been shown that legumain expression is inversely related to BDNF-TrkB activity. Here we show that legumain can conversely act on TrkB-BDNF by processing the C-terminal linker region of the TrkB ectodomain in vitro. Importantly, when in complex with BDNF, TrkB was not cleaved by legumain. Legumain-processed TrkB was still able to bind BDNF, suggesting a potential scavenger function of soluble TrkB towards BDNF. The work thus presents another mechanistic link explaining the reciprocal TrkB signaling and δ-secretase activity of legumain, with relevance for neurodegeneration.

## 1. Introduction

Cell signaling allows the transfer of information between cells to coordinate different tissues and cell types [1,2].

Of particular relevance are neuronal receptors such as the enzyme-coupled receptors p75NTR and Tyrosine-receptor-kinase B (TrkB), which transduce apoptosis and survival signals, respectively. These receptors work in concert with each other [3]. By these interactions, TrkB determines the neuronal fate concerning proliferation and survival; axonal and dendritic growth as well as remodeling; development of neuronal cytoskeleton; membrane traffic and synapse formation; and neuronal function and plasticity [4,5]. An important downstream effector protein of TrkB signaling is legumain (LEG). Legumain is a clan CD, family C13 cysteine protease [6], primarily located in the endo-lysosome [7]. There it fulfills important immunologic functions, e.g., producing immunopeptide for MHCII loading [7,8]. Legumain can also escape the lysosome primarily during pathophysiologic conditions like cancer and neurodegenerative diseases. There it is found to be active in the cytoplasm and extracellular space [9,10,11,12,13]. On the one hand, legumain’s expression is suppressed by TrkB signaling [14]. On the other hand, in the absence of TrkB/BDNF (Brain-derived neurotrophic factor) signaling, the transcription factor C/EBPβ is upregulated, resulting in increased legumain expression [14]. Furthermore, legumain-cleaved tau protein antagonizes TrkB signaling, [15].

Known physiologic regulators of TrkB are the neurotrophins BDNF and NT4, which specifically bind TrkB and induce its dimerization [5]. Additionally, TrkB is processed at its extracellular domain during excitotoxic events, whereby the exact identity of the involved sheddases remains unclear [16]. While sheddase activity is mostly mediated by membrane-anchored proteases, also soluble proteases contribute to shedding activity. Indeed, legumain was previously shown to act as δ-secretase on amyloid precursor protein (APP) processing without being membrane-bound [17,18].

Combining the above-mentioned information led us to explore the possibility of TrkB being processed by legumain in its ectodomain and its potential impact on TrkB binding to its specific binding partner BDNF. Here we identify legumain cleavage sites in the ectodomain of TrkB and show that this processing can create soluble TrkB isoforms with the ability to bind BDNF as well as the pro-form of BDNF. Additionally, we also show that the complex formation of TrkB and BDNF prevents the processing by legumain.

## 2. Results

### 2.1. Recombinant Protein Expression and Purification

To study the extracellular TrkB interaction with relevant partner molecules, we produced the ectodomain of TrkB (Ser29-Glu428) in HEK293S cells, carrying a C-terminal His6-tag for purification and concentration, Figure 1A,B. Additionally, we prepared the Ig2 domain of TrkB (His283-Glu428), which is proximal to the membrane and known to mediate neurotrophin binding [19,20,21]. We purified both proteins to apparent homogeneity with several glycosylation forms recognizable in the TrkB variants, Figure 1B and Appendix A. The identities of the proteins were further confirmed by HPLC-MS analysis. The Ig2 domain contains an unpaired cysteine residue, which led to partial dimerization under non-reducing conditions, as shown in Appendix A. This finding could provide a mechanistic explanation for the previously reported inactive dimer state of TrkB, which was shown to be mediated by the Ig2 domain [22]. To further assess the correct folding of the Ig2 domain, we confirmed that the correct Cys^302^-Cys^345^ is predominantly formed, Figure 1D. The alternative isoforms likely result from artificial disulfide shuffling during the course of sample analysis, as discussed by Lakbub et al. [23]. The purification of the HEK293S-produced TrkB variants resulted in 600 µg of pure protein from 30 mL and 60 mL medium for the ectodomain (ECD) and Ig2 variants of TrkB, respectively.

Additionally, we produced the neurotrophin BDNF in *E. coli* by in vitro folding from inclusion bodies, Figure 1C and Appendix A. The proteins were synthesized as preforms and matured by furin processing, Figure 1C and Appendix A. The identity and correct folding of BDNF was confirmed by mass spectrometric measurement, revealing the correct intact mass, Appendix A. The disulfide bonding pattern within BDNF is cysteine-1 (Cys^143^) with cysteine-4 (Cys^210^), cysteine-2 (Cys^188^) with cysteine-5 (Cys^239^), and cysteine-3 (Cys^199^) with cysteine-6 (Cys^241^). The first disulfide bond of Cys^143^-Cys^210^ could be directly detected, whereas the second and third bonds, Cys^188^-Cys^239^ and Cys^199^-Cys^241^, could not be separated but detected in a common peptidic fragment. Proteolytic cleavage proved impossible and reflected the sequential and spatial proximity of cysteine 5 and 6, which are located in the tripeptide Cys^239^-Val^240^-Cys^241^. The used protease, Trypsin, does not cleave after valine. The production protocol resulted in approximately 300 µg of pure protein from 1 g inclusion bodies for BDNF. Legumain was produced as previously described [24].

### 2.2. TrkB Processing by Legumain

We tested whether co-incubation of legumain and TrkB would lead to TrkB processing. Indeed, we could see approximately 10 kDa shifts of the TrkB ECD as well as the TrkB Ig2 band in the presence of legumain, Figure 2A. To test whether the fragment was cleaved from the N- or C-terminus of the TrkB protein variants, we immunoblotted the gel using an anti-His6 antibody and found that the C-terminal His6 tag was only present for the unprocessed TrkB variants, Figure 2B. Therefore, we conclude that the cleavage occurred C-terminally to the Ig2 domain. It is important to note that legumain, albeit a primarily lysosomal protease, retains significant activity also at neutral pH and in the absence of reducing agents, Appendix A.

### 2.3. Cleavage Site Analysis

We further determined the exact cleavage sites by mass spectrometry and found a dominant site at Asn^391^ and two minor sites at Asp^385^ and Asn^389^. The relative cleavage frequencies were pH dependent, consistent with the pH-dependent substrate preference of legumain, Figure 2C,D. Legumain cleaves after aspartate only at acidic conditions [24,25]. Importantly, Asn^389^Pro^390^Asn^391^ are strictly conserved in mammals, Appendix A. The cleavage sites are positioned after the Ig2 domain in the unstructured linker to the membrane, Figure 2D. The cleavage releases the ectodomain of TrkB from the membrane.

The structural model further suggests that the ectodomain structure, and presumably its function, remains intact after cleavage, Figure 2D. To test the structural integrity of the ectodomain, we analyzed its migration behavior on gel filtration chromatography. We found an increased retention volume of the legumain-processed as compared to the unprocessed TrkB variants, Appendix A, consistent with the structural model.

### 2.4. BDNF Binding Protects TrkB from Processing by Legumain

After having observed the legumain processing of the TrkB ectodomain, we next investigated whether the processing would be affected in the presence of the neurotrophin BDNF, known to bind TrkB at its Ig2 domain [19,20,21]. We, therefore, incubated the TrkB ECD and TrkB Ig2 with BDNF. In the lane without BDNF (−) and legumain (−), the TrkB ECD runs at the expected height (~65 kDa), with multiple bands representing the glycosylation isoforms and the additional cell culture-medium-derived BSA. In the presence of legumain (+), the full-length TrkB ECD shifts to a ~55 kDa band, revealing the BSA band at 60 kDa; the 36 kDa band represents legumain. Subsequently, we tested whether the pre-incubated TrkB neurotrophin complexes were still processed by legumain. We observed the appearance of additional degradation bands when legumain was added to TrkB ECD in the absence of BDNF. By contrast, when TrkB ECD was pre-incubated with the neurotrophin, no TrkB degradation bands were observed, Figure 3. The effect can be seen even clearer for TrkB Ig2. Upon addition of legumain to the TrkB Ig2, visible at approximately 25 kDa, the band shifted quantitatively by ~10 kDa, while this shift was suppressed completely in the presence of BDNF, Figure 3. In the presence of pro-BDNF, the observed suppression of TrkB processing was much less pronounced, Appendix A.

### 2.5. Binding Affinities of Soluble TrkB Variants to (Pro)BDNF

Given the impact of neurotrophins on the TrkB processing by legumain, we next investigated the impact of legumain processing of TrkB ectodomain variants on their binding affinities to BDNF. We used microthermophoresis to study the interaction of both partner molecules in the solution. For the unprocessed TrkB variants, we found comparable binding affinities of BDNF to the full-length ectodomain and the Ig2 with K_d_ values of 106 ± 44 nM and 67 ± 11 nM, respectively, Appendix A. Importantly, also the legumain-processed full-length ectodomain and Ig2 domain exhibited significant, yet two to four-fold reduced, affinities to BDNF with K_d_ values of 188 ± 82 nM and 300 ± 22 nM, respectively, Figure 4. Not surprisingly, the TrkB ectodomain variants revealed an approximately three to six-fold lower binding affinity to pro-BDNF as compared to mature BDNF, Figure 4C,D and Appendix A. Intriguingly, we observed secondary binding events with a micromolar affinity for BDNF but not for proBDNF with the TrkB variants (Figure 4A,B and Appendix A).

## 3. Discussion

### 3.1. Quality Control of Recombinant Proteins

We produced TrkB ectodomain and Ig2 domain in HEK293S cells to warrant proper posttranslational modifications and structural integrity. For BDNF, we took advantage of a previously reported preparation protocol in *E. coli* [26], with some modifications as specified in the Methods section. As a quality control, we determined intact masses and the correct disulfide bonding patterns for the recombinant proteins. In all cases, the expected intact masses and disulfide patterns were confirmed as the dominant isoforms, Figure 1D and Appendix A. Given that the mature BDNF (His^131^-Arg^249^) carries no glycosylation site, we thus established recombinant expression systems resembling native protein forms for TrkB and BDNF.

### 3.2. Legumain as a Potential Transient Secretase

We found that legumain is able to cleave the TrkB ectodomain after the Ig2 domain in the linker region to the transmembrane segment, Figure 1A. Importantly, the legumain cleavage site is strictly conserved in mammalian TrkB, Appendix A. Functionally, this cleavage represents a secretase activity of legumain on TrkB. Typically, secretases are membrane-anchored. While legumain is a soluble protein, it indeed can associate with membrane-bound receptors, including the integrin receptor α_V_β_3_ [25], as indicated in Figure 5. Legumain, thus, may represent a transient secretase co-localized with its substrate TrkB to the membrane for its selective shedding. By contrast, TrkB in complex with adjunct BDNF is protected from shedding by legumain, Figure 3. Structurally, this protective effect is likely caused by the interaction of the cleavage site with neurotrophins, competing with legumain shedding [27].

### 3.3. Legumain-Shed TrkB Binds BDNF

To evaluate possible physiological roles of the shed TrkB variants, we tested its binding towards BDNF and found an approximately 2–4 fold decreased binding affinity, Figure 4 and Appendix A, consistent with the partial release of the linker region, which was shown to contribute to neurotrophin binding [27]. On the other hand, the binding clearly shows that the structural integrity of legumain-processed TrkB is still intact, in line with the analytical gel filtration experiment, Appendix A.

### 3.4. Potential Functional Relevance of Shed TrkB

Our observation that legumain-shed TrkB is able to bind BDNF indicates a possible physiological regulatory role. Shed TrkB can bind and scavenge BDNF and thereby attenuate its signaling capacity, Figure 5. The relevance of BDNF scavenging by TrkB shedding in the context of stroke has been previously reported, worsening stroke-induced damage to neuronal cells [16]. Also, proBDNF exhibited significant TrkB binding, with approximately 3–6 fold weaker affinity than mature BDNF, consistent with earlier reports [28,29]. We speculate whether TrkB could thereby co-localize proBDNF with its activating protease furin to the membrane.

### 3.5. Biphasic Binding Behavior of Soluble TrkB-BDNF Complexes

Of note, we observed a biphasic binding behavior of the BDNF-TrkB complex in the full-length TrkB ectodomain variants as well as in the legumain-processed TrkB variants. This effect was observed in two experimental setups. In one, TrkB is fluorescently labeled and kept at constant concentration with BDNF increasing in concentration; in the other case, it is the reverse, i.e., BDNF is fluorescently labeled at constant concentration, and TrkB concentration increasing, see Figure 4 and Appendix A. This observation allows for several interpretations, including an additional BDNF binding site on TrkB with micromolar affinity. However, there are alternative binding events that might consistently explain the data. Based on our dynamics light scattering data and consistent with literature reports, we assume BDNF to be present as a dimer, Appendix A [5,30,31]. In the setup where TrkB is kept at constant concentration, and BDNF is increased in concentration, we expect to see BDNF-driven dimerization of TrkB. However, at increasingly high BDNF concentrations, BDNF dimers will be present at large excess to TrkB monomers, resulting in a saturation of TrkB monomers with BDNF dimers, effectively preventing TrkB dimerization, Appendix A. Excess of BDNF_2_ dimers will dilute and dissolve existing TrkB_2_-BDNF_2_ tetramers into TrkB_1_-BDNF_2_ trimers. Such a biphasic concentration-dependent receptor dimerization is well characterized in immunology, for instance, in rat basophile release assays [32].

We propose an alternative binding event to account for the biphasic behavior when BDNF is kept constant, and TrkB concentration is increased. Initially, TrkB monomer binding to a BDNF dimer will trigger TrkB dimerization because it energetically benefits from the bivalent interactions with TrkB and BDNF. At high TrkB concentrations, TrkB should be present as dimers almost exclusively, consistent with related structures from crystallographic experiments on TrkA, B and C and in vivo TrkB studies [22,33,34]. Consequently, TrkB dimers will bind to BDNF dimers partly also in a mono-valent manner, which allows for multimerization of the BDNF-TrkB system, Appendix A. Multimeric complexes tend to stick to capillary surfaces, which reduces the concentration of freely floating fluorescently labeled BDNF [35]. Indeed, during the MST measurement, the analysis software reported the adsorption of the fluorescently labeled protein, which can straightforwardly explain the biphasic behavior. The concentration dependence of the biphasic curve suggests a K_d_ for the soluble TrkB dimerization of approximately 1–5 µM.

Interestingly, we found that also the pro-form of BDNF binds to different TrkB ectodomain variants, albeit with approximately 3–5 times lower affinity, Figure 4. We assume that the observed binding event corresponds to the heteromolecular dimer formation of a proBDNF monomer with a TrkB monomer. As the dimerization of BDNF was a critical element to explain its biphasic binding behavior (Appendix A), we should expect no biphasic binding curve for monomeric proBDNF, which is indeed the case, Figure 4. We speculate whether the proBDNF-TrkB complex might support the activation of proBDNF, which is carried out by membrane-bound proteases like furin or plasmin [36]. The concentration of proBDNF at the membrane dramatically increases the likelihood of an enzyme-substrate encounter complex and its subsequent activation, Appendix A.

### 3.6. Conclusions

Many proteins were reported to exhibit moon-lightning activities in addition to function, which was initially discovered and attributed to them. Similarly, legumain can exhibit important functions outside the lysosome, its primary localization. When binding to receptors such as α_V_β_3_, legumain not only retains enzymatic activity at neutral pH over long times but also co-localizes to potential membrane-anchored substrates. Here, we provided in vitro evidence that legumain can function as a BDNF-dependent TrkB sheddase, which suppresses TrkB-medicated pro-survival signaling in the absence of BDNF.

## 4. Materials and Methods

### 4.1. Materials

The following chemicals were bought from Merck Millipore (Darmstadt, Germany): MgSO_4_, ammonium-acetate, citric acid, KCl, Na_2_HPO_4_, KH_2_PO_4_, HCl, and EDTA. The following chemicals were bought from Applichem (Darmstadt, Germany): Triton X100, Guanidine, Glutathione, HEPES, CaCl_2_, Urea, and NaCl. The following chemicals were purchased from Sigma-Aldrich (Vienna, Austria): Tris-Base, β-mercaptoethanol, L-Arginine, and formic acid. Tween-20 was obtained from Roth (Karlsruhe, Germany), acetonitrile from VWR Chemicals (Radnor, PA, USA) and imidazole from NeoLab Migge (Heidelberg, Germany). The pHLsec plasmids were kindly provided by A. R. Aricescu (Cambridge, UK).

### 4.2. Expression of TrkB Variants in HEK293S Cells and Purification

**Cloning.** The coding sequences for the TrkB variants TrkB ECD and TrkB Ig2 were based on the according NCBI RefSeq NM_012731.2 and synthesized (Eurofins Genomics, Munich, Germany). The sequences were cloned into a pHLsec plasmid with restriction digestion by Age1 and Nde1 (New England Biolabs, Ipswich, MA, USA) and T4 DNA Ligase ligation (Thermo Fisher Scientific, Vienna, Austria). The pHLsec plasmid also adds a C-terminal His6-tag to the TrkB variants. The correct insertion and orientation of the coding sequence for TrkB ECD and TrkB Ig2 were confirmed by sequencing (Eurofins Genomics, Germany).

**Expression.** Expression of the TrkB variants is based on the work of Aricescu et al. [37]. In short, adherent HEK293S cells were passaged and expanded in T-175 flasks (Greiner Bio-One, Kremsmünster, Austria) until ~90% confluency. Cells were maintained in DMEM medium (Sigma Aldrich, Vienna, Austria) supplemented with 10% fetal bovine serum (Gibco, Carlsbad, CA, USA), non-essential amino acids (Sigma Aldrich, Vienna, Austria) and L-glutamine (Sigma Aldrich, Vienna, Austria) at 37 °C, 5% CO_2_. 5 mL serum-free medium was mixed with 85 µg of TrkB plasmid DNA and 175 µL PEI stock (1 mg/mL) to allow DNA-PEI complex formation. Transfection was achieved by adding the mix to the confluent cells in 30 mL 2% FBS medium, briefly shaking the flask and putting it back into the incubator for 24 h. After that, the 2% FBS medium was removed and replaced by a 30 mL serum-free medium. Medium with secreted soluble TrkB variants was harvested after 72 h.

**Purification.** Harvested medium (30 mL per T-175 flask) was diluted 1:5 with wash buffer (50 mM Tris, 300 mM NaCl, pH 7.5) and incubated together with 3 mL Ni^2+^-NTA (Qiagen, Hilden, Germany) at 4 °C for 60 min. The flow-through was separated from the beads, and the beads were washed 2 times with 10 mL wash buffer. TrkB variants were eluted with 2–4 times 3 mL elution buffer (50 mM Tris, 300 mM NaCl, 250 mM Imidazole, pH 7.5). The elutions were pooled and concentrated with a 3 kDa Amicon (Merck Millipore, Darmstadt, Germany) to ~4 mg/mL.

### 4.3. Expression of Neurotrophin BDNF in E. coli as Inclusion Bodies, Folding and Purification

**Cloning.** The coding sequence for proBDNF was taken from the ENA database (#AAA63483) and synthesized (Eurofins Genomics, Germany). The sequences were cloned into a pET22b plasmid (Merck Millipore, Darmstadt, Germany) with restriction digestion by Xho1 and Nde1 (New England Biolabs, Ipswich, MA, USA) and T4 DNA Ligase ligation. The pET22b plasmid also adds a C-terminal His6-tag to proBDNF. The correct insertion and orientation of the coding sequence for proBDNF were confirmed by sequencing (Eurofins Genomics, Germany).

**Expression.** Rosetta 2 (DE3) *E. coli* (Merck Millipore, Darmstadt, Germany) were transformed with the plasmid and grown as a multiclonal culture in 50 mL standard LB medium (Roth, Karlsruhe, Germany) with 100 µg/mL Ampicillin and 20 µg/mL Chloramphenicol at 37 °C (both Applichem, Darmstadt, Germany). 600 mL LB-Ampicilin medium was inoculated with 3–4 mL of overnight culture. These cultures were grown at 37 °C to an OD600nm of 0.8–1.0. Expression was induced with 1 mM IPTG (ForMedium, Norfolk, UK), and the culture was further incubated at 37 °C for 3–4 h under continued shaking. Cells were harvested at 4000× *g* for 10 min at 4 °C. Pellets of 3–4 harvests were pooled and stored at −20 °C.

**Inclusion Body Preparation.** The pooled pellet was resuspended in at least 30 mL wash buffer 1 (50 mM Tris, 500 mM NaCl, 20 mM EDTA, pH 8.0) and sonicated (Bandelin, Berlin, Germany) at near 100% power, 50% duration, 9 intervals per 3 min on ice. A spatula tip of DNase (Applichem, Darmstadt, Germany) and 5 mM MgSO4 was added and incubated for 30 min at 4 °C on a rolling incubator. 2% Triton X-100 was added and manually shaken until the Triton was dissolved. The resuspended pellet was centrifuged at 17,500× *g* for 15 min at 4 °C, and the supernatant was discarded. Next, the pellet was washed 2 times with 4 mL per gram pellet wash buffer 2 (50 mM Tris, 500 mM NaCl, 20 mM EDTA, pH 8.0, 2% Triton X-100) followed by 2 washes with wash buffer 1. Between each wash step, the sample was centrifuged at 17,500× *g* for 15 min at 4 °C and resuspended with a Potter homogenizer. The final pellet was weight. Per gram pellet, 10 mL inclusion body solubilization buffer 1 (6 M guanidine-HCL, 50 mM Tris, 20 mM EDTA, 100 mM β-mercaptoethanol, pH 8.5) was added and stirred until the pellet was dissolved. After that, 2 M HCl was added until a pH of approximately 3.5 was reached (pH strips). The solution was centrifuged at 17,500× *g* for 30 min at RT. Next, the supernatant was transferred into a 10 kDa cutoff dialysis membrane (Repligen, DG Breda, The Netherlands) and dialyzed overnight against 2 L dialysis buffer 1 (50 mM NaCl, 10 mM EDTA, pH 4.5). The dialysis buffer was changed 2 times during this time. The dialysis product was centrifuged at 17,500× *g* for 30 min at RT, and the supernatant was discarded. The resulting inclusion body pellet is dissolved in 4–6 mL per gram pellet solubilization buffer 2 (6 M guanidine-HCl, 50 mM Tris, 20 mM EDTA, pH 3.5) followed by centrifugation at 17,500× *g* for 20 min at RT. The supernatant was transferred into a new 15 mL tube, and the concentration was measured with a Bradford Assay [38]. Preferred concentration for (re-) folding: 15–20 mg/mL or more.

**Folding.** Folding of proBDNF from inclusion bodies was done by rapid dilution. For this, 50% of the inclusion body supernatant was dripped into 100 times the total supernatant volume of 4 °C cold folding buffer (0.75 M L-Arginine, 100 mM Tris, 1 mM EDTA, pH 9.0, GSH:GSSG 10:1) under constant slow stirring. After 4 h, the remaining 50% was slowly added, and stirring was continued for at least 4 more hours or overnight. Next, the folding solution was concentrated on ice with a peristaltic 5 kDa filter pump (Heidolph Instruments, Schwabach, Germany) and then dialyzed (Nadir membrane, Roth, Karlsruhe, Germany) against 5 L dialysis buffer 2 (20 mM HEPES, 100 mM NaCl, pH 7.0, 4 °C and RT). For thorough removal of guanidine, arginine, Tris, GSH, and GSSG, dialysis was carried out as follows: First dialysis for 2 h at RT, second dialysis overnight at 4 °C, last for 2 h at RT. For each dialysis step, the membrane tube containing the protein was shortly opened and mixed by pipetting and then transferred into a fresh dialysis buffer. After that, the dialysis product was centrifuged at 17,500× *g* for 15–30 min at 4 °C. The supernatant was transferred into a fitting tube or flask, and the concentration was measured with Bradford Assay.

**proBDNF Purification.** Purification of folded proBDNF was achieved by SP-sepharose cation exchange chromatography (Cytiva Lifesciences, Vienna, Austria). Approximately 2.5 mL of SP-sepharose beads were equilibrated with equilibration buffer (20 mM HEPES, 50 mM NaCl, pH 7.0) at 4 °C and transferred into the folding dialysis supernatant under constant slow stirring for 60 min. After, the suspension was transferred back into the column to collect the sepharose beads with bound proBDNF. At least 2 wash steps (20 mM HEPES, 100 mM NaCl, pH 7.0) with 4 times the column volume were carried out. Elution was achieved by adding 2.5 mL of elution buffer (20 mM HEPES, 500 mM NaCl, pH 7.0) and incubation on a roller incubator (CAT, Ballrechten-Dottingen, Germany) for 10 min at 4 °C. Elution steps were repeated at least 4 times. The concentration of each elution was measured with Bradford Assay. To decrease the high salt concentration of the elution buffer, the samples were rebuffered into the wash buffer over an NAP column (GE Healthcare, Vienna, Austria). The concentration was again measured with Bradford Assay to determine losses.

**Activation.** The rebuffered elutions were pooled, and 5 mM CaCl2 and hFurin in a ratio of 1:100 (*m*/*m*) were added. The mixture was incubated on a roller incubator for at least 3.5–4 h at RT. To stop the activation of proBDNF by Furin, 10 mM EDTA (pH 7.4) was added, and the mixture was centrifuged at 17,500× *g* for 15 min at 4 °C.

**BDNF Purification.** The activation assay was immediately loaded on a 1 mL SP-sepharose cation exchange column and incubated for 60 min at 4 °C. Washing, Elution and NAP-rebuffering of BDNF was carried out as described during proBDNF purification, except that the elution buffer contained 1 M salt instead of just 500 mM and the buffer for rebuffering was 1x PBS pH 7.4.

### 4.4. HPLC-MS Analysis for TrkB Ig2 and BDNF

**General HPLC-MS parameters**. For all chromatographic analyses, mobile phase A was H_2_O + 0.10% formic acid, and mobile phase B was acetonitrile + 0.10% formic acid. Chromatographic separations for intact Ig2, legumain-digested Ig2, peptide and disulfide mapping of Ig2 as well as intact proBDNF were carried out on a Discovery BIO wide pore C18 column (150 × 2.1 mm i.d., 3.0 μm particle size, 300 Å pore size, Supelco, Bellefonte, PE, USA). While queued for analysis, samples were stored in the autosampler at 4.0 °C. The mass spectrometers were mass calibrated using Pierce™ LTQ Velos ESI Positive Ion Calibration Solution from Life Technologies (Vienna, Austria).

**Analysis of intact Ig2, legumain-digested Ig2, peptide and disulfide mapping of Ig2**. For the MS analysis, Ig2 samples were precipitated by using ice-cold methanol and resuspended using 150 mM ammonium acetate. Trypsin was added in an enzyme-protein ratio of 1:20 and allowed to digest the protein at 37 °C overnight at 900 rpm. The reaction was stopped by the addition of 10% formic acid to a final concentration of 1%. Chromatographic separation was carried out on a capillary HPLC instrument (UltiMate™ U3000 RSLC, Thermo Fisher Scientific, Germering, Germany). Ten microliters of intact Ig2 and legumain-digested Ig2 samples [0.25 mg.mL^−1^] were injected using in-line split-loop mode and separated at a flow rate of 200 µL.min^−1^ and a column oven temperature of 70 °C applying the following linear gradient: 5.0% B for 10.0 min, 5.0–40.0% B for 20 min, 40.0–80.0% B for 5.0 min, 80.0% B for 5.0 min and 5.0% B for 20 min. UV detection was carried out at 214 nm using a 1.4 µL flow cell.

All samples were analyzed on a Thermo Scientific™ QExactive™ benchtop quadrupole-Orbitrap^®^ mass spectrometer equipped with an Ion Max™ source with heated electrospray ionization (HESI) probe, both from Thermo Fisher Scientific (Bremen, Germany), and an MXT715-000-MX Series II Switching Valve (IDEX Health & Science LLC, Oak Harbor, WA, USA). Mass spectrometric data were acquired between minute 10.0 and 55.0 in an *m*/*z* range of 500–3000 with instrument settings as specified earlier [39]. Intact Ig2 was measured at a resolution of 17,500 at *m*/*z* 200, and legumain-digested Ig2 was analyzed at a resolution of 140,000 at *m*/*z* 200. MS/MS of intact and legumain-digested Ig2 was carried out applying all ion fragmentation (AIF) in the higher-energy collisional dissociation (HCD) cell at normalized collision energy (NCE) settings of 22.0 within a scan range of *m*/*z* 400–2500 and a resolution setting of 140,000 at *m*/*z* 200. Peptide and disulfide mapping was conducted using data-dependent MS/MS. Each scan cycle consisted of a full scan at a scan range of *m*/*z* 400–2000 with an AGC target of 3 × 10^6^, a maximum injection time of 100 ms and a resolution setting of 70,000, followed by 5 data-dependent HCD scans at 30 NCE with an AGC target of 5 × 10^5^, a maximum injection time of 200 ms and a resolution setting of 17,500. The dynamic exclusion was set to 10.0 s.

Isotopically resolved mass spectra were deconvoluted using the Xtract algorithm implemented in the Xcalibur™ software version 3.0.63 (Thermo Fisher Scientific, Waltham, MA, USA). Isotopically unresolved spectra were deconvoluted using the Respect algorithm implemented in Biopharma Finder 1.0 (Thermo Fisher Scientific, Waltham, MA, USA). AIF data were evaluated using the software ProSight Lite v1.4 Build 1.4.6 provided by the Kelleher Research Group (Northwestern University, Evanston, IL, USA); mass tolerance for annotation of b- and y-fragments was set to 25 ppm. Peptide mapping data were evaluated using Biopharma Finder 1.0.

**Analysis of proBDNF and tryptic peptides thereof**. For intact mass analysis, 300 µg of TCA precipitated proBDNF were dissolved in 175.0 mmol L^−1^ ammonium acetate. For disulfide mapping, 400 µg of TCA-precipitated proBDNF were dissolved in 100 µL 175.0 mmol L 1 ammonium acetate. Trypsin (1:20 enzyme protein ratio, Promega, Mannheim, Germany) was directly added and allowed to digest overnight at 37 °C and 900 rpm. The reaction was stopped by the addition of FA to a final concentration of 1%.

Intact proBDNF analysis was carried out in a Vanquish HPLC system (Thermo Fisher Scientific, Germering, Germany). Five microlitres of the sample were injected and separated during a linear gradient at 60.0 °C and a constant flow rate of 100.0 µL min^−1^: 15.0% B for 3.0 min, 15.0–90.0% B for 27.0 min, 90.0–99.0% B for 0.1 min, 99.0% B for 5.0 min, 99.0–5.0% B for 0.1 min, 5.0% B for 5.0 min.

Tryptic peptides were analyzed on a nano-HPLC system (UltiMate™ U3000 RSLCnano, Thermo Fisher Scientific, Germering, Germany) with an autosampler set to 4.0 °C and column oven to 50 °C. proBDNF-derived peptides were separated on an Acclaim PepMap^®^ RSLC column (15 cm × 300 µm i.d., 2.0 µm particle size, 100 Å pore size, Thermo Scientific™). One microlitre of the sample was injected in full-loop mode. proBDNF-peptides were separated at 1.2 µL min 1. For peptide analyses, the linear gradient was as follows: 1.0% B for 5.0 min, 1.0–30% B for 30.0 min, 30.0–60.0% B for 5.0 min, 99.0% B for 5.0 min, 1.0% B for 10.0 min.

Intact proBDNF, as well as tryptic peptides thereof, were analyzed on a benchtop Q Exactive™ Plus Hybrid Quadrupole-Orbitrap™ Mass Spectrometer (Thermo Fisher Scientific, Bremen, Germany). Intact proteins were sprayed from a HESI source in positive ion mode. Ion source settings for proBDNF were as follows: source heater temperature 150 °C, spray voltage 4.5 kV, sheath, auxiliary, and spare gas flow 10, 8, and 0 arbitrary units, respectively. proBDNF was analyzed in full MS mode throughout the chromatographic run (0.0–50.0 min) over an *m*/*z* range of 1000–3500 at a resolution of 140,000 at *m*/*z* 200 using the following instrument settings: AGC target 3 × 10^6^, maximum injection time of 150 ms, 10 microscans, capillary temperature 300 °C, and S-lens RF level 50. Tryptic peptides of proBDNF were ionized on a nano-ESI source under positive polarity at 1.5 kV. Peptides were acquired in full MS mode throughout the chromatographic run (0.0–55.0 min) within a scan range of *m*/*z* 400–3000 at a resolution of 70,000 at *m*/*z* 200 with an AGC of 3 × 10^6^, maximum injection time of 100 ms. The capillary temperature was set to 250 °C, S-lens RF level to 60. For fragmentation, a top 10 method was utilized within a scan range of 200–2000 *m*/*z* at a resolution of 17,500 at 200 *m*/*z* with an AGC of 1 × 10^5^ and a maximum injection time of 50 ms. The normalized collision energy was set to 28.0% and the dynamic exclusion to 10.0 s. All used HPLC-MS parameters are summarized in Appendix A.

For disulfide mapping, simulated isotope patterns were generated in Thermo Xcalibur Qual Browser software (v.4.2.28.14) with the following settings: the chemical formula was calculated in excel, 100% hydrogen adduct, most abundant charge state 2–4, output style profile (Gaussian, samples per peak 70), resolution 70,000, valley FWHM (full-width half maximum). Extracted ion current chromatograms were calculated by Thermo Xcalibur Qual Browser software using the following settings: enabled smoothing using Boxcar for 7 points, baseline subtraction enabled, mass tolerance 10 ppm. Peptide data was evaluated in Byonics (Protein Metrics Inc., Cupertino CA, USA). For proBDNF-peptides, lysine and arginine were defined as cleavage sites allowing 2 missed cleavages. Parameters for instrument settings were precursor mass tolerance of 20 ppm, fragmentation type CID low energy, and a fragment mass tolerance of 20 ppm. The protein false discovery rate (FDR) was set to 1%. Spectrum output options: charge state 1–6, maximum precursor mass 10,000 Da; protein output options: 2% FDR. Disulfide mapping was activated (−2.015650 Da at cysteine) was activated for proBDNF. The mass spectra of intact proBDNF were deconvoluted using BioPharma Finder 3.0 (Thermo Scientific) using the Xtract^®^ algorithm for isotopically resolved spectra. Detailed settings for evaluation parameters can be found in Appendix A.

### 4.5. Legumain Fluorogenic Activity Assay Site

The enzymatic activity of legumain was investigated using the peptidic Z-Ala-Ala-fluorescence was measured at 460 nm upon excitation at 380 nm. Processing Asn-7-amino-4-methylcoumarin (Z-AAN-AMC; Bachem, Bubendorf, Switzerland) substrate. The activity was measured in assay buffers composed of 100 mM citric acid pH 5.5 and 50 mM NaCl or 100 mM Hepes pH 7.0 and 50 mM NaCl supplemented with 50 µM of the substrate. Additionally, the effect of the reducing agent DTT was tested upon the addition of 5 mM DTT to the assay buffers. Reactions were started by the addition of the enzyme at an 8 nM concentration. Assays were carried out at 37 °C in an infinite M200 plate reader (Tecan, Grödig, Austria). An increase in fluorescence was measured at 460 nm upon excitation at 380 nm.

### 4.6. Cleavage Assay to Determine Cleavage Site

Processing of TrkB ECD and Ig2 with legumain was done by mixing 0.5–1 mg/mL TrkB ECD/Ig2 and 0.007–0.055 mg/mL legumain in cleavage assay buffer (20 mM citric acid, 50 mM NaCl, pH 5.5) which was pH-fixed to pH 7 or 5.5 with either 1 M HEPES, 1 M Tris pH 7 or 1 M citric acid pH 5.5 (1/10th of total volume). The mixture was incubated for 10–15 min at RT. For MS measurements, legumain was inactivated by adding 1 M Ammonium acetate pH 8.6 at 1:1 ratio of total volume. An exemplary assay with respective ratios and volumes can be found in Appendix A.

### 4.7. Confirmation of Structural Integrity of TrkB after Processing by Gel Filtration

The TrkB cleavage assay was done as described above and shown in the Appendix A. Only the total volume has been up-scaled to 500 µL.

Processed TrkB ECD was injected into the AKTA FPLC system (GE Healthcare, Vienna, Austria) with a Superdex 200 10/300 GL column (Cytiva Lifesciences) attached, while for TrkB Ig2, a Superdex 75 10/300 GL was used. Both columns were equilibrated with 1x PBS pH 7.4. Fractions with no visible legumain on SDS-PAGE were pooled and concentrated with a 5 kDa cutoff Amicon (Sartorius, Göttingen, Germany).

### 4.8. BDNF-TrkB Complex Cleavage Assay

**Preparation**. To allow complex formation, 47 µL of 0.3 mg/mL BDNF were mixed with 3 µL of ~4 mg/mL TrkB ECD/Ig2 for a total concentration of ~0.28 mg/mL of both in 1x PBS pH 7.4. The mixture was incubated at RT for 10 min. Legumain was diluted in cleavage assay buffer to a stock concentration of 1.875 mg/mL.

**Cleavage assay**. The complex cleavage assay was then mixed together similarly to the cleavage assay described before or as described in Appendix A and incubated at RT for 10 min. The stock solution for fixing the pH to 7 was 1 M Tris, and the assay buffer was the previously described cleavage assay buffer at pH 5.5. 10 µL samples were transferred onto a 15% SDS-PAGE.

### 4.9. Determination of Complex Formation Affinity by Microscale Thermophoresis (MST)

**Protein labeling**. BDNF (20 µM), intact TrkB ECD/Ig2 (20 µM/20 µM), processed TrkB ECD/Ig2 (4.9 µM/15 µM) were labeled with NT-647 RED dye (NanoTemper, Munich, Germany) according to the protocol provided by NanoTemper with the following change: The buffer was not exchanged with the protocols exchange buffer, but all proteins were kept in either 1x PBS pH 7.4 or 20 mM HEPES, 100 mM NaCl, pH 7.5.

**Incubation**. For the following combinations of labeled and unlabeled proteins, each preparation step was the same: labeled intact TrkB ECD/Ig2 (0.65 µM/0.79 µM) with unlabeled BDNF (20.5 µM), labeled processed TrkB ECD/Ig2 (0.25 µM/0.52 µM) with unlabeled BDNF (20.5 µM) and labeled BDNF (1.1 µM) with unlabeled TrkB ECD/Ig2 (20 µM/20 µM). Absorption at 280 and 650 nm was measured for each labeled protein to determine the concentration and degree-of-labeling (DOL) before being diluted 1:67 or 1:100 in 180 µL ice cold MST buffer (1x PBS, pH 7.4, 0.005% Tween 20). Next, 10 µL of unlabeled protein was diluted 1:2 per step in a 16x dilution series with MST buffer at RT. To each 10 µL of the dilution series, 10 µL of diluted labeled protein was added and mixed by pipetting. The mix was incubated in the dark at RT for 10 min and centrifuged for 5 min at 13,000 rpm at 4 °C (Heraeus biofuge fresco, Hanau, Germany). Thereafter, each of the 16 dilution steps was loaded into a capillary (Monolith NT.115, Nanotemper, Munich, Germany) and placed into the NT.115-Monolith (NanoTemper, Munich, Germany) thermophoresis device.

**Measurement and data generation**. MST signals were measured with the Monolith NT.115 MST device and recorded with MO.Control Software (v1.6.1, NanoTemper). Excitation Power was set to 40% for Nano-RED, and MST-Power was set to medium. Acquired data were evaluated and affinities were determined with MO.Affinity Software (v2.3, NanoTemper).

## Figures and Tables

**Figure 1 ijms-24-05394-f001:**
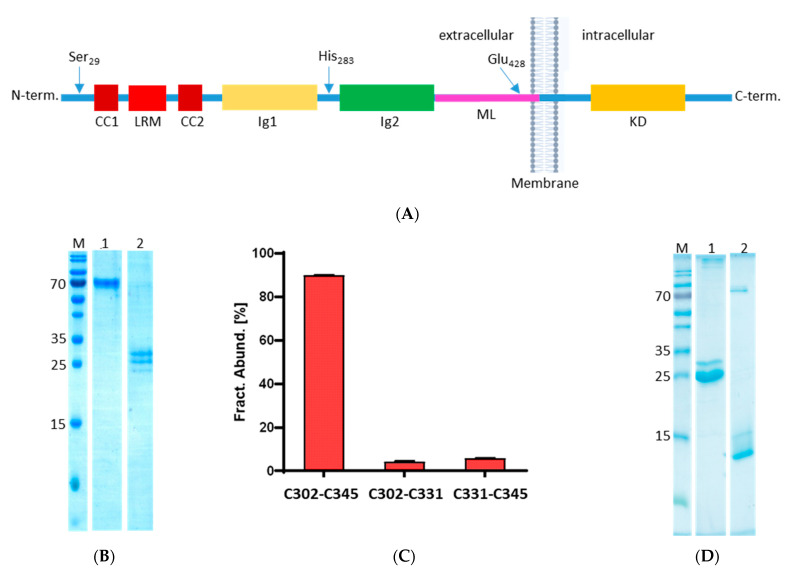
Preparation of rat TrkB and proBDNF. (**A**) Domain organization of TrkB. CC1/2: cysteine cluster 1/2; LRM: leucine-rich-motif; Ig1/2: Ig-like domain 1/2; ML: membrane linker; KD: kinase domain. (**B**) SDS-PAGE of TrkB variants. Lane 1: purified TrkB ECD; Lane 2: purified TrkB Ig2 domain with different glycosylation isoforms. (**C**) HPLC-MS analysis of the relative fractional abundance of the correct disulfide isoform Cys302-Cys345 in TrkB Ig2. Fractional abundances of the disulfide bonds, as detected by mass spectrometry, are normalized frequencies, i.e., they add up to 100%. (**D**) SDS-PAGE of (pro-)BDNF. Lane 1: purified proBDNF; Lane 2: furin-activated BDNF; the 70 kDa band corresponds to inactivated furin.

**Figure 2 ijms-24-05394-f002:**
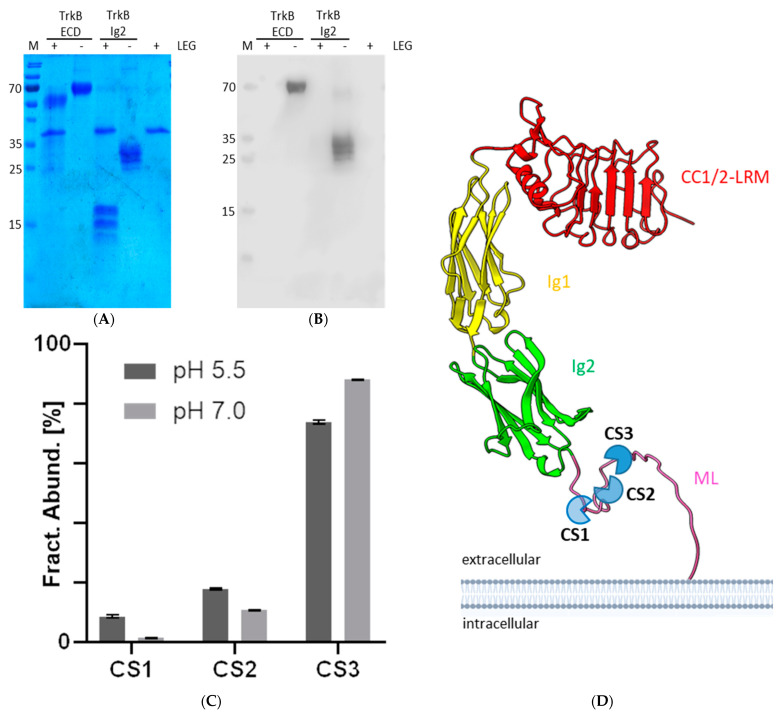
Legumain processes TrkB. A+B: Coomassie-stained SDS PAGE gel (**A**) and corresponding Immunoblot (**B**) of TrkB ECD and Ig2 domain in the presence (+) and absence (−) of legumain, denoted as LEG. An anti-His6 antibody is used in the Immunoblot to detect the C-terminal His-tag of the uncleaved TrkB variants. M represents the molecular marker lane. (**C**) Relative fractional abundance of three detected cleavage sites CS1 (Asp385), CS2 (Asn389), and CS3 (Asn391) at pH 5.5 and 7.0. Fractional abundances of the individual cleavage events, as detected by mass spectrometry, are normalized frequencies, i.e., they add up to 100%. (**D**) Cartoon representation of the TrkB ECD with the three detected cleavage sites CS1, CS2, and CS3 (Asp385, Asn389, and Asn391) in the membrane linker segment with their cleavage predominance indicated by their color intensity.

**Figure 3 ijms-24-05394-f003:**
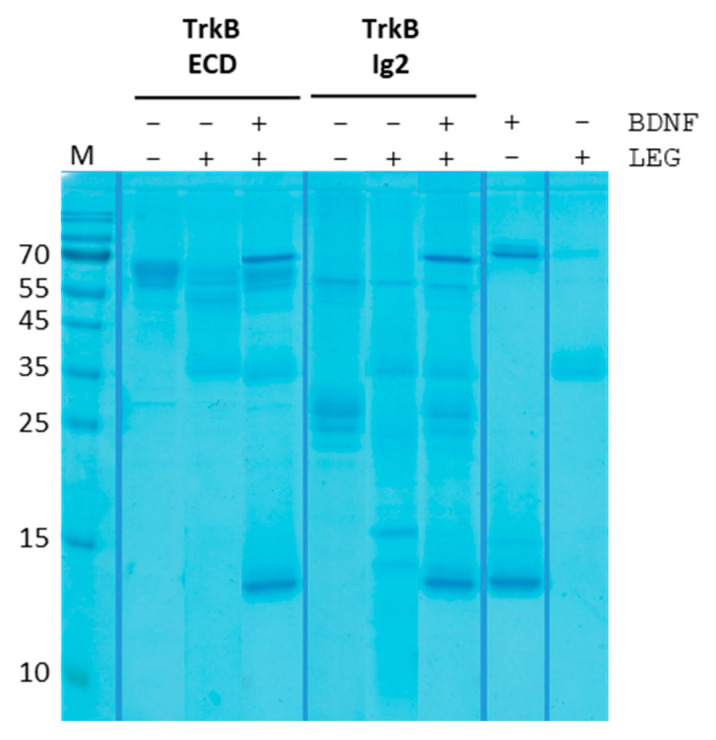
TrkB in complex with BDNF is protected from processing by legumain. M represents the protein marker. In the next three lanes, the presence or absence (+/−) of BDNF and legumain (“LEG”) on the processing of TrkB ECD is tested, followed by three lanes on the processing of TrkB Ig2 domain. The outer right lanes represent controls of BDNF and legumain (LEG) only, migrating at approximately 14 kDa and 36 kDa, respectively. Note the presence of cell medium-derived BSA migrating at 60 kDa. Additionally, in the BDNF sample, inactivated furin is visible at 70 kDa, which was used to activate proBDNF.

**Figure 4 ijms-24-05394-f004:**
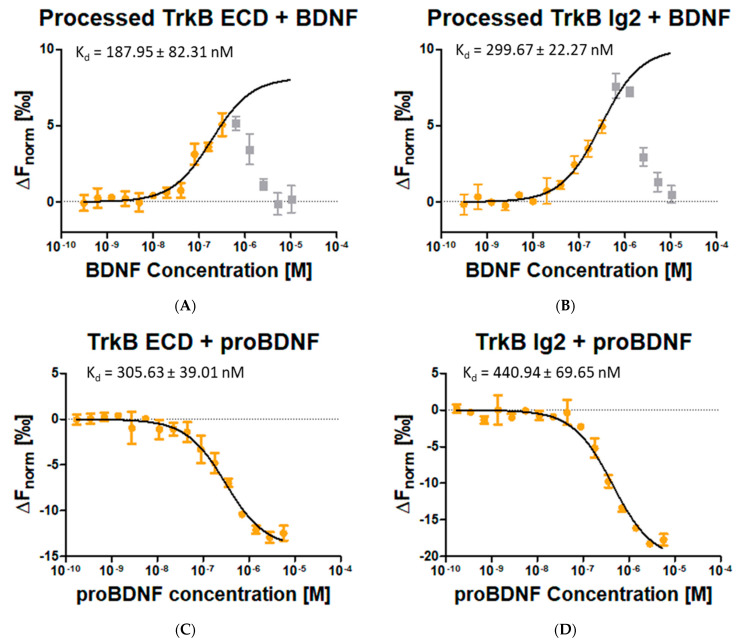
Legumain-processed TrkB variants bind BDNF. Non-processed TrkB variants bind proBDNF. (**A**) The legumain-processed TrkB ECD binds BDNF in solution with a K_d_ of 188 nM. (**B**) The legumain-processed TrkB Ig2 binds BDNF in solution with a K_d_ of 300 nM. Of note, in both cases, secondary binding events at micromolar concentrations were observed. (**C**) TrkB ECD binds proBDNF in solution with a K_d_ of 306 nM. (**D**) TrkB Ig2 binds proBDNF in solution with a K_d_ of 441 nM. Contrasting the situation with mature BDNF, no secondary binding events were observed.

**Figure 5 ijms-24-05394-f005:**
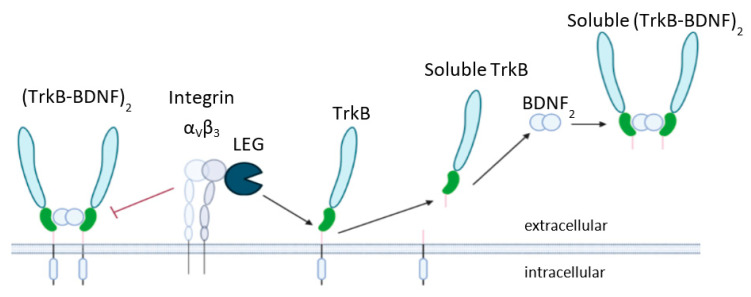
Possible impact of TrkB shedding by Legumain. On the left side, BDNF mediates a heterotetrameric complex with membrane-bound TrkB (TrkB-BDNF)_2_, which prevents shedding by legumain (“LEG”). In the absence of BDNF, TrkB can be shed from the membrane. Soluble TrkB can still bind BDNF, resulting in a soluble (TrkB-BDNF)_2_ complex.

## Data Availability

The data presented in this study are available on request from the first (Christoph Holzner) and corresponding author (Hans Brandstetter).

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
