# Peer review of "Legumain Functions as a Transient TrkB Sheddase"

_ijms, 2023, doi:10.3390/ijms24065394_

Round 1
Reviewer 1 Report
In the manuscript by Holzner C et al., the authors show that the cysteine protease legumain might cleave the neuronal TrkB receptor without abolishing the receptor binding capacity of BDNF. This is a new function of legumain not previously described. Although the topic is novel and interesting for the protease scientific community, the presentation and discussion of the data needs improvement as addressed below.
Comments and suggestions for the authors:
Some concerns that need to be clarified, discussed or are missing:
1. Data is missing showing legumain activity in the incubates/buffers of pH 5.5 and 7. Is legumain active in the buffers used and without the addition of any reducing agent? Table S4 shows the content in the cleavage assay at pH 7 but legumain activity in the buffer is not shown. This could easily be shown by legumain cleavage of a peptide substrate (like Z-Ala-Ala-Asn-AMC) in the buffers (pH 5.5 and 7) and needs to be shown, at least in the Supplementary.
2. The discussion is very fragmented in small chapters, mainly repeating the results and referring to the figures.
· Extracellular active legumain needs to be discussed and especially in context of pH.
· A conclusion is missing.
· Fig. 6 needs to be referred in the body text.
· Line 174: The reference 26 and 27 should be used in the Discussion section when discussing the binding affinity of BDNF and not when describing own results in the Result section.
· Line 221: “The relevance of BDNF scavenging by TrkB shedding in the context of stroke has been previously reported”. What context has been reported? – protects or enhances the risk of stroke?
· Line 251: What are “related structures”? – Examples should be included.
3. “May function” in the title is kind of “weak”. Do the authors believe that TrkB is cleaved and shedded by legumain or not? – remove “may”?
4. The abstract should define that legumain and δ-secretase is actually the same protease.
5. First paragraph of the introduction is very general concerning membrane receptor and should be omitted.
6. In the whole manuscript, legumain or LEG is used on-off. Since a number of names and functions of legumain is used by the scientific community (AEP, ACP, ligase, δ-secretase), the protease should be referred as legumain and not an abbreviation (LEG) all through the paper. This also not to confuse with the legumain gene name (LGMN).
7. In all figures:
· Some figure legends explains the experiment/analyses used, whereas other just refer the results. Preferentially, all figure legend should give the reader an understanding of the experiment and analyses performed, whereas the results should be explained in the body text of the Result section.
· Numbering A, B etc should be bold in upper left corner of each figure.
· The lane M on gels/blots should be explained in the figure legends (Molecular marker).
8. Fig. 1: Figure C and D should switch places, as the results in D (TrkB) is explained in the body text before C (BDNF).
9. Line 108 reads “We tested whether co-incubation of legumain and TrkB would lead to its processing.” What is “its”? – can be either legumain or TrkB? Needs to be rephrased.
10. Fig. 2:
· Figure A should be divided in A and B and the text could read “Coomassie-stained SDS PAGE gel (A) and corresponding Western immunoblot (B) of TrkB ECD and Ig2 domain…..”. Immunoblot (not Western) is used in the body text and should be reflected in the figure legends as suggested.
· B: What is Fract Abund. (%) on the Y-axis? – should be explained in the fig. legend.
· C: Why is CS1 purple, whereas CS2 and CS3 are blue? Since CS3 (Asn391) is the dominant cleavage site, it is suggested that the CS colors indicate this. All abbreviations should be explained in the legend (LRM, ML).
11. Fig. 3: The cleavage of TrkB ECD is not very convincing. What is the “expected height” (line 152; use molecular size) of TrkB ECD and why is BSA present in the incubate? – this is not easy to follow from the M&M section. The content of the two right lanes is not explained but might be the pure proteins? – should be explained in the legend.
12. Fig. 4 and 5 should be shown as one figure (A-D) and with a proper legend describing the experiment and analyses.

Author Response
REVIEWER #1
Some concerns that need to be clarified, discussed or are missing:
- Data is missing showing legumain activity in the incubates/buffers of pH 5.5 and 7. Is legumain active in the buffers used and without the addition of any reducing agent? Table S4 shows the content in the cleavage assay at pH 7 but legumain activity in the buffer is not shown. This could easily be shown by legumain cleavage of a peptide substrate (like Z-Ala-Ala-Asn-AMC) in the buffers (pH 5.5 and 7) and needs to be shown, at least in the Supplementary.
The requested additional enzymatic data are provided in supplementary figure S3A and show that legumain is indeed active in the absence of reducing agents at both pH values.
- The discussion is very fragmented in small chapters, mainly repeating the results and referring to the figures.
- Extracellular active legumain needs to be discussed and especially in context of pH.
Connecting with the newly added figure S3a, we clarified that legumain retains proteolytic activity at neutral pH and under non-reducing conditions, as present in the extracellular space. Additionally, we expanded the schematic figure 6, illustrating that extracellular ligands such as αVβ3 integrin can stabilize legumain in the extracellular environment at neutral pH.
- A conclusion is missing.
We have added a conclusion, as suggested by the reviewer.
- 6 needs to be referred in the body text.
We had a reference to figure 6 already in the discussion, line 220, section 3.4 Potential functional relevance of shed TrkB. Additionally, we have expanded figure 6 to include the transient anchoring and stabilization of legumain by extracellular receptors, allowing us to refer to figure 6 more prominently.
- Line 174: The reference 26 and 27 should be used in the Discussion section when discussing the binding affinity of BDNF and not when describing own results in the Result section.
We are thankful for this comment, the references are indeed misplaced
- Line 221: “The relevance of BDNF scavenging by TrkB shedding in the context of stroke has been previously reported”. What context has been reported? – protects or enhances the risk of stroke?
We clarified in the text that the BDNF scavenging tends to worsen the course of strokes.
- Line 251: What are “related structures”? – Examples should be included.
We clarify that the dimer formation has been described by the neurotrophin binding Ig2 domains of TrkA, TrkB, and TrkC.
- “May function” in the title is kind of “weak”. Do the authors believe that TrkB is cleaved and shedded by legumain or not? – remove “may”?
We thank the reviewer for the suggestion and clarified the statement, as requested.
- The abstract should define that legumain and δ-secretase is actually the same protease.
We clarified the legumain nomenclature as suggested.
- First paragraph of the introduction is very general concerning membrane receptor and should be omitted.
We condensed the first paragraph to a single sentence to avoid general text book contents.
- In the whole manuscript, legumain or LEG is used on-off. Since a number of names and functions of legumain is used by the scientific community (AEP, ACP, ligase, δ-secretase), the protease should be referred as legumain and not an abbreviation (LEG) all through the paper. This also not to confuse with the legumain gene name (LGMN).
We agree with reviewer 1 that the legumain nomenclature is indeed confusing and therefore use “legumain” throughout the text.
- In all figures:
- Some figure legends explains the experiment/analyses used, whereas other just refer the results. Preferentially, all figure legend should give the reader an understanding of the experiment and analyses performed, whereas the results should be explained in the body text of the Result section.
We thank the reviewer for this valuable suggestion. We kept the figure legends to clarifying the experiments, eliminating explanations of the results in the legends, which is now exclusively provided in the text.
- Numbering A, B etc should be bold in upper left corner of each figure.
We understand that the reviewer prefers the conventional labelling style. However, the figure panel labels A, B, C, … are placed automatically by the journal format. Consequently, we cannot change the positioning of the labels.
- The lane M on gels/blots should be explained in the figure legends (Molecular marker).
We explained the marker lane, as requested.
- Fig. 1: Figure C and D should switch places, as the results in D (TrkB) is explained in the body text before C (BDNF).
We appreciate this suggestion and implemented it accordingly.
- Line 108 reads “We tested whether co-incubation of legumain and TrkB would lead to its processing.” What is “its”? – can be either legumain or TrkB? Needs to be rephrased.
We agree that this statement requires clarification. We rephrased the sentence to “We tested whether co-incubation of legumain and TrkB would lead to TrkB processing.”
- Fig. 2:
- Figure A should be divided in A and B and the text could read “Coomassie-stained SDS PAGE gel (A) and corresponding Western immunoblot (B) of TrkB ECD and Ig2 domain…..”. Immunoblot (not Western) is used in the body text and should be reflected in the figure legends as suggested.
We divided the old panel A into two separate sub-panels, A and B, as suggested. Furthermore, we use Immunoblot consistently in the figure legend and text body.
- B: What is Fract Abund. (%) on the Y-axis? – should be explained in the fig. legend.
We clarified the meaning of fractional abundances in the context of figure 2 B (now figure 2C) as well as in figure 1D (now figure 1C). Briefly, the fractional abundances are defined such that the sum of the individual cleavage events, as detected by mass spectrometry, add up to 100 %.
- C: Why is CS1 purple, whereas CS2 and CS3 are blue? Since CS3 (Asn391) is the dominant cleavage site, it is suggested that the CS colors indicate this. All abbreviations should be explained in the legend (LRM, ML).
We agree that the representation may be misleading. We like the suggestion to code the predominance of the cleavage sites with the intensities of their colors.
- Fig. 3: The cleavage of TrkB ECD is not very convincing. What is the “expected height” (line 152; use molecular size) of TrkB ECD and why is BSA present in the incubate? – this is not easy to follow from the M&M section. The content of the two right lanes is not explained but might be the pure proteins? – should be explained in the legend.
Another excellent point. We now state that the TrkB ECD band shifts from ~65 kDa to ~55 kDa. Additionally, we clarify that BSA is present in the cell culture media, consistent with the description in the M&Ms.
- Fig. 4 and 5 should be shown as one figure (A-D) and with a proper legend describing the experiment and analyses.
We agree that the fusion of the two figures into one figure is more consistent and have implemented this change accordingly.

Reviewer 2 Report
The manuscript by Holzner and colleagues from the group of Hans Brandstetter from the University of Salzburg reports on results highlighting the interplay between the C13-family cysteine peptidase legumain and tyrosine-receptor-kinase B (TrkB) in the context of the neurotrophin, brain-derived neurotrophic factor (BNDF), which is a physiologic TrkB regulator that induces kinase dimerization. The authors found that the TrkB ectodomain is released by legumain, which can act as a soluble sheddase to liberate cell surface receptors and proteins, as previously shown for amyloid precursor protein. Shed TrkB can bind BNDF as well as its precursor. Moreover, BNDF binding to TrkB prevents its processing by legumain. Overall, the work has been carried out very carefully, as usual for the Brandstetter laboratory, and is suitable for publication.
Minor issue:
-Generally speaking, if the authors want to obtain less glycosylated proteins, an option is to use kifunensine during expression in HEK cells.
-In Figs. 4 and 5, the experimental errors preclude a precision of the Kd values to the hundredths place. I’d suggest precision-based rounding to 188 ± 82 nM, 300 ± 22 nM, 306 ± 39 nM, 441 ± 70 nM, etc.
Author Response
We are very happy to learn that the reviewer appreciated our manuscript and are very grateful for their constructive suggestions for future experiments.
Round 2
Reviewer 1 Report
Most of the concerns are now included in the revised manuscript by Holzner C et al.
The missing data of legumain activity at pH 5.5 and 7 are now included in the supplementary and method section.
The abbreviation LEG still needs to be explained in various figure legends (Fig. 2 and 6, Suppl. Fig. S2, S3), as well as in Table S4.
Author Response
Dear editor:
Thank you very much for your mail from 7 March 2023 regarding your decision on our manuscript “Legumain functions as a transient TrkB sheddase” (ijms-2247533). We are very happy to see that the reviewers appreciate the additionally included data of legumain activity at pH 5.5 and 7.
The only remaining request is
-
The abbreviation LEG still needs to be explained in various figure legends (Fig. 2 and 6, Suppl. Fig. S2, S3), as well as in Table S4.
We explained the abbreviation LEG in all figures and table S4, as requested.
We sincerely thank the reviewer for their constructive comments, which have significantly improved the quality of the manuscript. We feel that the manuscript is now suitable for publication.
With many thanks for handling the paper,
Hans Brandstetter